# Effects of Site, Genotype and Subsequent Harvest Rotation on Willow Productivity

**Mariusz Jerzy Stolarski \***[ID]**, Michał Krzyżaniak**[ID]**, Dariusz Załuski**[ID]**, Józef Tworkowski and Stefan Szczukowski**

Department of Plant Breeding and Seed Production, Faculty of Environmental Management and Agriculture, Centre for Bioeconomy and Renewable Energies, University of Warmia and Mazury in Olsztyn (UWM), Plac Łódzki 3, 10-724 Olsztyn, Poland; michal.krzyzaniak@uwm.edu.pl (M.K.); dariusz.zaluski@uwm.edu.pl (D.Z.); jozef.tworkowski@uwm.edu.pl (J.T.); stefan.szczukowski@uwm.edu.pl (S.S.)
* Correspondence: mariusz.stolarski@uwm.edu.pl; Tel.: +48-89-5234838

**Abstract:** Perennial crops harvested in short rotations provide substantial amounts of biomass. This study determined the survival rate, biometric features and yield of fresh and dry biomass of 15 willow genotypes (including seven varieties and eight clones), cultivated at two different sites in two consecutive three-year harvest rotations. The study revealed the very high impact of the genotype (81% of the total variance) on the willow yield. The harvest rotation, along with the genotype, had a significant impact on the plant survival rate and the number of shoots per stool. Willow biomass was mainly affected by the plant height, its survival rate and shoot diameter. The significantly highest fresh (106 Mg ha$^{-1}$) and dry biomass yield (54.0 Mg ha$^{-1}$) was obtained from the Żubr variety of *S. viminalis*, which distinguished this variety from the other genotypes. The mean yield for the best three and five genotypes was 13% and 17% lower, respectively, and the mean yield for the whole experiment was 37% lower compared to the mean yield of the best variety (Żubr). Therefore, the choice of a willow genotype is of key importance for successful willow production.

**Keywords:** Salix; genoype × site interaction; survivability; biometric features; plant height; fresh biomass yield; dry biomass yield

## 1. Introduction

Perennial crops, grown on agricultural land and harvested in short rotations, can provide substantial amounts of biomass [1–4]. Until now, biomass was usually used as energy feedstock and its use for industrial purposes and in integrated biorefineries has only recently been contemplated [5–8].

There are three major categories of perennial crops grown for biomass. These are short rotation woody crops (SRWCs) such as willow, poplar, eucalyptus, black locust; herbaceous crops such as Virginia mallow, willow-leaf sunflower, cup plant; grasses such as giant miscanthus, prairie cordgrass, switchgrass and giant reed [9–11]. Although such a large diversity of perennial crops offers the advantage of providing farmers with many choices, one has to possess sufficient knowledge in this regard. Obtaining a high biomass yield throughout the period of the plantation use (which has a significant impact on the production process profitability) requires making a proper selection of species and cultivars suitable for specific weather and site conditions [12–14].

Willow has several advantages as an SRWC, including a wide range of genetic diversity, easy reproduction, tolerance to a wide range of site conditions, ability of new shoots to rapidly regrow after multiple harvests, possibility of harvests in different rotations, resistance to disadvantageous environmental conditions, e.g., morning frost, wet snow and strong winds [4,15,16]. Therefore,

willow biomass production in short harvest rotations has been researched in many countries, including Europe [17–19], Canada and the USA [20–23]. Production of willow as an SRWC can be profitable [24,25], its energy balance is positive [26,27] and it brings measurable environmental benefits [15,28–30].

The advantages of willow and benefits from its cultivation as an SRWC are very important and can encourage potential producers to decide to start willow biomass production. However, for willow production to be economically, energy-efficiently and environmentally viable, it has to provide stable and high biomass yield per unit area [31]. The main factors that determine the willow biomass yield have been analysed for years and they include: (1) the choice of a suitable species and cultivar [18,23,31,32]; (2) soil conditions [13,18]; (3) weather conditions [31,33]; (4) agrotechnical procedures, including the fertilisation type and rate [17,34]; (5) planting density and harvesting frequency [9,31]. It should be noted that all of these factors are important, and they have a combined impact on the final effect, i.e., the willow biomass yield. However, when attempting to rank these factors, the choice of the species and variety should be identified as the most important. In consequence, if a biomass producer chooses a wrong willow species or variety, he will not achieve a high biomass yield even if the plantation is set up at a very good site, with the optimum weather and agrotechnical conditions. Moreover, the biomass producer, who has a very limited influence (or sometimes none) on the site or weather conditions, enjoys the full (100%) responsibility for the choice of plant species and variety.

Therefore, because of the large diversity of willow species, varieties and genotypes fit for cultivation as SRWCs and since willow production produces higher yield (per year of plantation use) in three-year harvest rotations [4,9,31], research was conducted to determine (1) survival rate and biometric features; (2) yield of fresh and dry biomass of 15 willow genotypes (including seven varieties and eight clones), cultivated at two different sites in two consecutive three-year harvest rotations. These findings were subsequently used to: (3) quantitatively determine the relative contribution of genetic and site-related factors and their interactions in explaining the variance of willow survival rate, biometric features and biomass yield.

## 2. Materials and Methods

### 2.1. Field Experiments

Two identical single-factorial field experiments were conducted in 2013–2018 at two experimental stations of the University of Warmia and Mazury (UWM), located near the villages of Bałdy and Obory in Poland. The experiment at Bałdy (Warmińsko-Mazurskie Province, 53°35′48″ N, 20°36′12″ E) was set up on mud-muck soil developed on calcareous gyttja in loamy subsoil. Willow cuttings were planted in April 2008 at a high density of 48 thousand per ha. The cuttings were planted in two-row strips with rows spaced every 0.75 m. The strips were spaced every 0.90 m. The cuttings in each row were planted every 0.25 m. Willows were coppiced in 2008–2012 in one-year rotations because it was believed at the time that it was a potentially interesting method for willow cultivation in small farms. However, as the market situation has changed since 2013, it was decided to extend the willow harvest cycle, which is why the plants were harvested twice in two subsequent three-year rotations: the first one (I) covering the years 2013–2015, and the second (II) the years 2016–2018. An identical field experiment was set up in April 2009 at Obory (Pomorskie Province, (53°43′34″ N, 18°53′55″ E) on humic heavy alluvial soil, formed from silty clay. The planting method was the same as at the Bałdy site. The willow at Obory was also coppiced in one-year rotations in 2009–2012. Subsequently (like at Bałdy), starting with 2013, willow was harvested twice in consecutive three-year rotations (I—2013–2015, II—2016–2018). Therefore, it must be explained that although the Bałdy experiment was set up a year earlier (2008) than the Obory experiment (2009), it had no significant direct impact on the results analysis in later three-year harvest rotations (2013–2018). This was a consequence of the fact that both at Bałdy and Obory, willows had been harvested earlier every year: five and four times, respectively. Plants harvested in the first three-year rotation grew on eight- and seven-year old stools, so the root

systems can be regarded as very well-developed and the plants were giving the full yield. The stools in the second three-year rotation were 10 and 11 years old.

Fifteen identical willow genotypes of nine different species and interspecies hybrids were tested at each site (Bałdy and Obory) (Table 1). The 15 willow genotypes included seven varieties registered at the Centre for Cultivar Testing in Słupia Wielka, and eight clones—all of them bred and kept at the collection of UWM. At each site (Bałdy and Obory), each genotype was planted in a 150-m² strip. At both sites, within each strip, three 40-m² plots were randomly designated to conduct biometric measurements and determine the plant density and biomass yield.

The same mineral fertilisation was applied in each experiment before the next three-year rotation was begun. Therefore, the following fertilisers were applied in April 2013 and 2016: nitrogen as ammonium nitrate (90 kg ha⁻¹ N), phosphorus as triple superphosphate (13 kg ha⁻¹ P) and potassium—as potassium salt (50 kg ha⁻¹ K).

**Table 1.** Genotypes (varieties and clones) of willow tested in two identical field experiments at Bałdy and Obory.

| Species | Genotype Status (Variety or Clone) | Name |
| --- | --- | --- |
| *Salix viminalis* | variety | Start |
| *S. viminalis* | variety | Sprint |
| *S. viminalis* | variety | Turbo |
| *S. viminalis* | variety | Tur |
| *S. viminalis* | variety | Kortur |
| *S. viminalis* | variety | Oltur |
| *S. viminalis* | variety | Żubr |
| *S. acutifolia* | clone | UWM 093 |
| *S. alba* | clone | UWM 095 |
| *S. dasyclados* | clone | UWM 155 |
| *S. fragilis* | clone | UWM 195 |
| *S. pentandra* | clone | UWM 035 |
| *S. triandra* | clone | UWM 198 |
| *S. viminalis* × *S. amygdalina* | clone | UWM 054 |
| *S. viminalis* × *S. purpurea* | clone | UWM 033 |

*2.2. Willow Survival Rate, Biometric Features and Biomass Yield*

The plant density on three plots at each site for each harvest rotation was determined for each genotype after the growing seasons ended and it was subsequently calculated per 1 ha. Moreover, the plant survival rate (%) was determined. The height (m) and diameter (mm) of shoots were measured in ten randomly selected plants on each plot. The height was measured from the ground level to the top of the tallest plant shoot. The shoot diameter was measured 0.50 m above the ground level. The number of shoots per stool was also determined in ten replicates on each plot. Only live shoots taller than 1.50 m were taken into account, whereas dry shoots were excluded. Thus, each analysed biometric feature (height, diameter, number of shoots) was measured 30 times for each genotype at each site and for each harvest rotation. This makes up a total of 1800 measurements of each feature, which constituted input data for further analyses.

The plants were harvested in winter (February 2016 and 2017) manually with a chainsaw after each three-year rotation. Immediately after harvest, the whole mass of shoots from each plot was weighed (within an accuracy of 0.01 kg) and the fresh biomass yield was calculated (Mg ha⁻¹ f.m.). Representative biomass samples (approximately 3 kg) were collected during harvest from entire plants of each genotype from each plot to determine the moisture content in it. The biomass moisture content was determined at 105 °C using the oven dry method (EN ISO 18134-1:2015). Subsequently, the moisture content and the fresh biomass yield were used to calculate the dry biomass yield (Mg ha ⁻¹ d.m. (dry matter yield)). The biomass yields from the three-year rotations were divided by three to present the biomass yield per one year of plantation use. Moreover, a ranking of the mean dry biomass yield

(Mg ha$^{-1}$ year$^{-1}$ d.m.) for 15 willow genotypes from two sites in two consecutive three-year rotations was developed.

### 2.3. Statistical Analysis

All statistical analyses were conducted with STATISTICA 13.3 software (TIBCO Software Inc., Palo Alto, CA, USA 2017). Such variables as survival rate, number of shoots per stool, shoot diameter, plant height, fresh matter yield (f.m.) and dry matter yield (d.m.) were analysed statistically by a repeated-measures ANOVA, with site and genotype as the grouping factors and the replicates were nested in the site effect. The rotation effect was the only factor used for repeated measures. The significance of the factors and their interactions were tested at the significance level of $\alpha = 0.05$. The statistics $F$ from these analyses are shown in the table of results. The percentage share of all the analysis effects under study in the total sum square (total SS) was calculated. This measure explained the share of individual factors in the trait variance. Moreover, Tukey's honest significant difference (HSD) test with $p < 0.05$ was used to evaluate the significance of differences between the subsequently determined means and homogeneous groups. Multiple regression analysis was applied in an assessment of the relationship between fresh and dry matter yield and the morphological features of plants. The variability of the analysed features in relation to the experimental conditions was evaluated using coefficients of variation (CV%).

### 2.4. Soil and Weather Conditions during the Experiments

Soil examinations at Bałdy showed that the mud-muck soil was neutral, with pH 7.2 in KCl. The humus content was 16.3%. The content of $P_2O_5$, $K_2O$ and Mg was: 1435 mg, 560 mg and 637 mg kg$^{-1}$ of soil, respectively. The complete humic heavy alluvial soil at Obory was also neutral with pH 7.0 in KCl. The humus content was lower: 7.09%. The content of $P_2O_5$, $K_2O$ and Mg was also lower than at Bałdy: 488 mg, 300 mg and 157 mg kg$^{-1}$ of soil, respectively. However, despite the differences in the contents of these elements, the soil at both sites (Bałdy and Obory) was good and fertile.

The sites were about 150 km apart, which is why the weather conditions were different. The average air temperature during six growing seasons (April–October) at Obory ranged from 13.3 °C to 16.0 °C, in 2017 and 2018, respectively (Table S1). The average temperature at Obory was always higher compared to Bałdy by 0.1 °C to 1.3 °C. Therefore, the annual average air temperature during the six years of the experiment at Obory (9.0 °C) was higher by approximately 0.7 °C compared to Bałdy. July and August were always the warmest months, with average monthly temperatures reaching 21 °C. The highest temperatures at both sites during the six growing seasons of the experiment were recorded in 2018.

It was the opposite case with precipitation because the average precipitation over six years at Bałdy (598 mm) was higher by approximately 20% than at Obory (Figure S1). The average precipitation during the growing season (April–October) at Bałdy (414 mm) was approximately 18% higher than Obory. Moreover, particularly large differences in the precipitation between the study sites were recorded during the second three-year harvest rotation (2016–2018) because precipitation during all the three growing seasons was higher at Bałdy than at Obory by 80–159 mm. The largest difference was recorded during the hottest growing season of 2018, when 419 mm of precipitation was recorded at Bałdy and only 260 mm at Obory. On the other hand, the highest precipitation during the whole six-year study period was recorded during the growing season of 2017 (533 and 612 mm, at Obory and Bałdy, respectively). The annual precipitation was also the highest in 2017: 640 and 803 mm, respectively.

## 3. Results

### 3.1. Plant Survival Rate, Number of Shoots, Plant Height and Shoot Diameter

The willow survival rate, number of shoots per stool and plant height differed significantly depending on the site, genotype, harvest rotation and interactions between most of these factors

(Table 2). The shoot diameter was significantly differentiated only by the genotype, harvest rotation and the genotype-site interaction. The mean plant survival rate for all genotypes, sites and harvest rotations was 42.0% after the ten years of the experiment (Figure 1). Notably, since the initial planting density was very high (48 thousand per ha), the plant density in the last three-year rotation was still high—approximately 20 thousand per ha. The willow survival rate was differentiated to the greatest extent by the genotype—61% of the total variance, followed by the harvest rotation (17%) and the genotype–site interaction (13%) (Table 2). Among the genotypes under study, the highest survival rate was observed in the hybrid *S. viminalis* × *S. amygdalina* UWM 054 (nearly 62%) and the lowest was UWM 093 of *S. acutifolia* (only 20%) (Figure 1). The mean survival rate at Obory was higher by 1.5 percentage point (pp) compared to that at Bałdy. The mean willow survival rate in the first harvest rotation was 46.4%, whereas it decreased significantly in the second rotation by 8.8 pp.

The genotype had the greatest share in the variance of the number of shoots per stool (29%), followed by the harvest rotation, site and their interaction (Table 2). Among the genotypes under study, the significantly highest mean number of shoots (5.6) per stool was found on *S. fragilis* UWM 195, and the fewest were on UWM 093 of *S. acutifolia* (Figure 2). The mean number of shoots per stool at Bałdy (4.6) was larger by one than at Obory. It was similar when this attribute in the second harvest rotation was compared to the first rotation. It should be linked to the lower plant survival rate at Bałdy than at Obory and to decreasing plant survival rate in the second harvest rotation compared to the first harvest rotation. Owing to the smaller plant density, a larger number of shoots per rootstock were able to grow and survive.

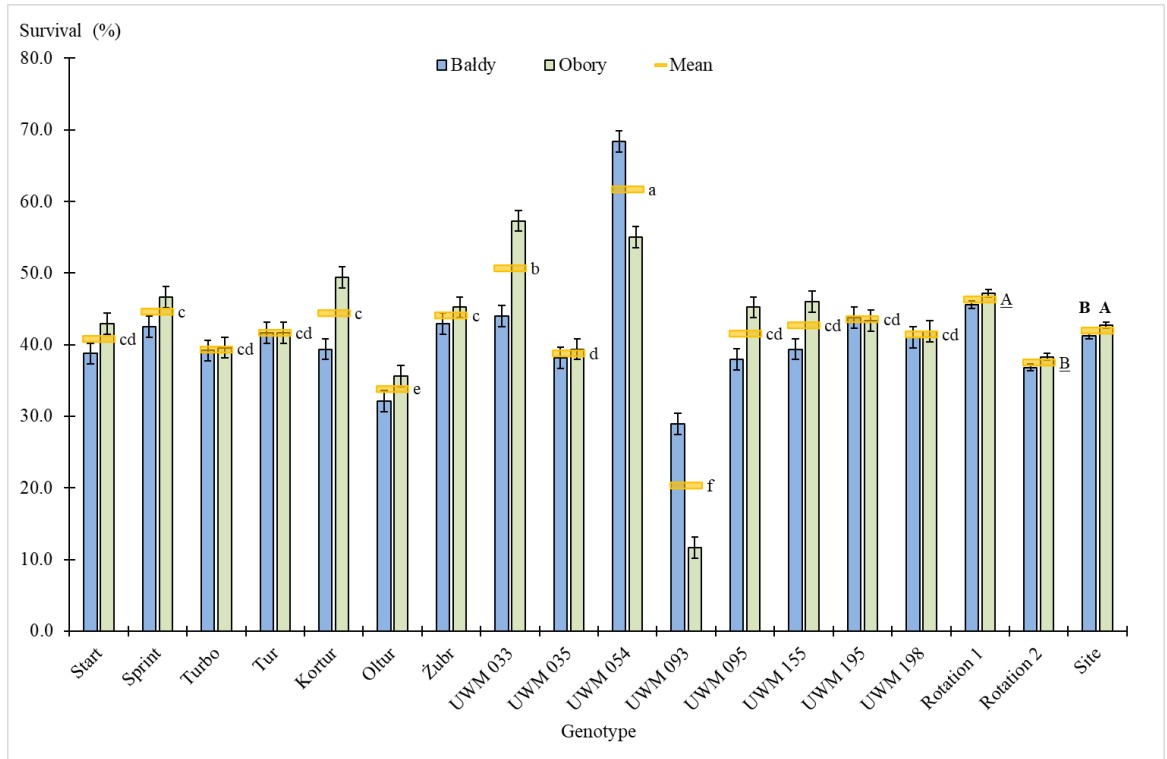

**Figure 1.** The survival rate of 15 willow genotypes from two sites in two consecutive three-year rotations (legend: error bars denote the standard error of mean; lower case letters a–f denote homogeneous groups from the Tukey test for mean values for genotypes regardless of the site and show differences between genotypes; underlined upper case letters A, B denote homogeneous groups for mean values for harvest rotation and show differences between rotations; bold upper case letters A, B denote homogeneous groups for mean values for sites and show differences between sites; ns denote not significant).

**Table 2.** Statistics *F* from the repeated measure variance analysis and the percentage share of effects in the total sum of squares of an attribute.

| Source of Variation | df | Survival | | No. of Shoots | | Height | | Shoot Diameter | | Fresh Matter Yield | | Dry Matter Yield | |
|---|---|---|---|---|---|---|---|---|---|---|---|---|---|
| | | *F* | Share (%) | *F* | Share (%) | *F* | Share (%) | *F* | Share (%) | *F* | Share (%) | *F* | Share (%) |
| Site | 1 | 8.1 ** | 0.5 | 68.4 *** | 9.2 | 35.3 *** | 1.6 | 2.3 | 0.5 | 0.9 | 0.0 | 1.8 | 0.1 |
| Rep (Site) | 4 | 1.70 | 0.4 | 3.3 * | 1.8 | 0.7 | 0.1 | 0.3 | 0.2 | 0.4 | 0.0 | 0.4 | 0.0 |
| Genotype (Gen) | 14 | 68.4 *** | 60.7 | 15.7 *** | 29.3 | 131.3 *** | 80.9 | 18.9 *** | 53.3 | 196.5 *** | 81.2 | 194.4 *** | 81.1 |
| Site × Gen | 14 | 14.4 *** | 12.8 | 5.6*** | 10.5 | 13.0 *** | 8.0 | 5.0 *** | 14.1 | 20.2 *** | 8.3 | 20.0 *** | 8.3 |
| Error 1 | 56 | | 3.6 | | 7.5 | | 2.5 | | 11.3 | | 1.7 | | 1.7 |
| Rotation (Rot) | 1 | 425.1 *** | 17.0 | 88.8 *** | 15.5 | 26.2 *** | 1.2 | 14.7 *** | 3.0 | 136.3 *** | 3.7 | 133.9 *** | 3.7 |
| Rot × Site | 1 | 0.00 | 0.0 | 6.1 * | 1.1 | 9.9 ** | 0.4 | 0.1 | 0.0 | 15.5 *** | 0.4 | 19.5 *** | 0.5 |
| Rot × Rep (Site) | 4 | 1.20 | 0.2 | 0.9 | 0.6 | 1.4 | 0.3 | 0.5 | 0.4 | 0.4 | 0.0 | 0.4 | 0.0 |
| Rot × Gen | 14 | 1.70 | 0.9 | 3.6 *** | 8.8 | 3.2 *** | 2.0 | 0.8 | 2.3 | 4.6 *** | 1.8 | 3.8 *** | 1.5 |
| Rot × Site × Gen | 14 | 2.8 ** | 1.6 | 2.5 ** | 6.1 | 1.0 | 0.6 | 1.1 | 3.3 | 3.4 *** | 1.3 | 3.7 *** | 1.4 |
| Error 2 | 56 | | 2.2 | | 9.7 | | 2.5 | | 11.6 | | 1.5 | | 1.6 |
| Total | | | 100.0 | | 100.0 | | 100.0 | | 100.0 | | 100.0 | | 100.0 |

Share (%) percentage share in the total sum of squares; * *p* < 0.05; ** *p* < 0.01; *** *p* < 0.001.

The plant height was determined to the greatest extent by the genotype (nearly 81%), followed by interaction of the site and genotype (8%) (Table 2). Among the 15 genotypes under study, plants of the Żubr variety (*S. viminalis*) were the significantly tallest (6.9 m), whereas those of the UWM 093 genotype of *S. acutifolia* were the shortest (4.1 m) (Figure 3). Plants at Bałdy were taller by 0.2 m on average compared to those growing at Obory and the plants were taller in the second rotation.

An assessment of the impact of the studied factors on the shoot diameter showed similar dependence for the plant height, as this attribute is determined the most strongly by the genotype (53%), followed by the interaction of site and genotype (14%) (Table 2). Shoots of the Żubr variety (*S. viminalis*) were also the thickest: mean 35.5 mm. On the other hand, shoots in the UWM 093 genotype and the Start variety were the thinnest (Figure 4). It is also noteworthy that individual variance in the objects under study had a greater share compared to the other shoot diameter attributes and the number of shoots per stool.

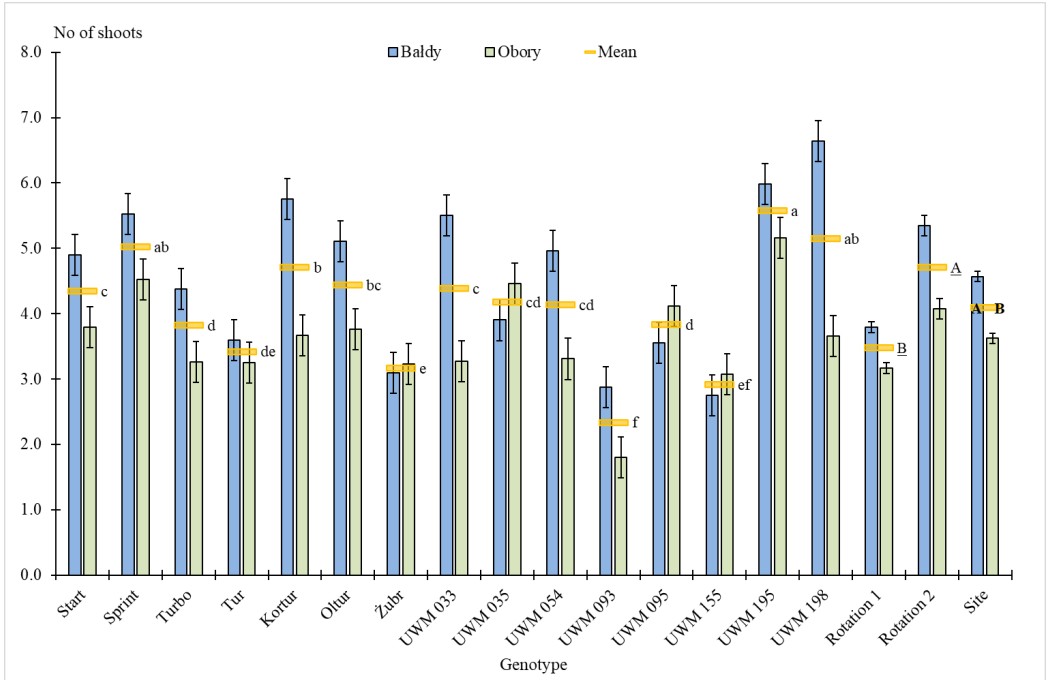

**Figure 2.** The number of shoots per stool of 15 willow genotypes from two sites in two consecutive three-year rotations (legend: see Figure 1).

### 3.2. Biomass Yield

Both the fresh and dry biomass yield were differentiated by the genotype and harvest rotation and by their interaction. However, no site impact on the biomass yield was observed (Table 2). The biomass yield was differentiated to the greatest extent by the genotype (81% of the total variance), followed by the interaction of genotype and site (8%) and harvest rotation (4%). Among the genotypes under study, the significantly highest fresh biomass yield (106 Mg ha$^{-1}$) was obtained from the Żubr variety of *S. viminalis*, which distinguished this variety against the other genotypes (Figure 5). On the other hand, because of the lowest survival rate and the poorest biometric features, the fresh biomass yield for the UWM 093 genotype of *S. acutifolia* was very low (8 Mg ha$^{-1}$), which practically eliminates this genotype from further research devoted to willow yield assessment. Furthermore, the mean yield of the other 13 genotypes was lower from 17% up to 54%, for *S. fragilis* UWM 195 and Start, respectively, when compared with the Żubr variety. The fresh biomass yield in the first three-year harvest rotation, when the plants were harvested from the 7-year-old stool, was higher by over 9 Mg ha$^{-1}$ compared to the second rotation yield.

When the biomass moisture content was taken into account, the relationships for dry biomass yield were similar to those for the fresh biomass yield. The significantly highest mean dry biomass yield (54.0 Mg ha$^{-1}$) was obtained from the Żubr variety of species *S. viminalis* (Figure 6). On the other hand, the poorest UWM 093 genotype of species *S. acutifolia* yielded only 4 Mg ha$^{-1}$. The other 13 genotypes gave a lower yield, from 17% up to 56%, for UWM 195—an *S. fragilis* genotype—and UWM 155 of *S. dasyclados*, respectively. The mean dry biomass yield in the first three-year harvest rotation (36.6 Mg ha$^{-1}$) was higher by 4.8 Mg ha$^{-1}$ compared to the second rotation yield.

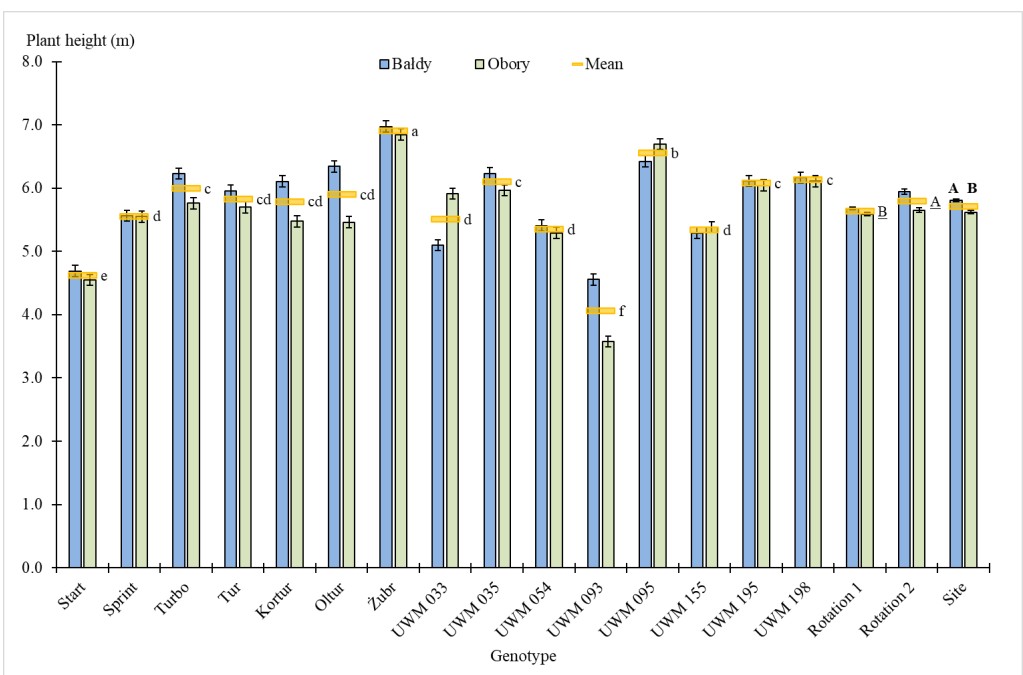

**Figure 3.** The plant height of 15 willow genotypes from two sites in two consecutive three-year rotations (legend: see Figure 1).

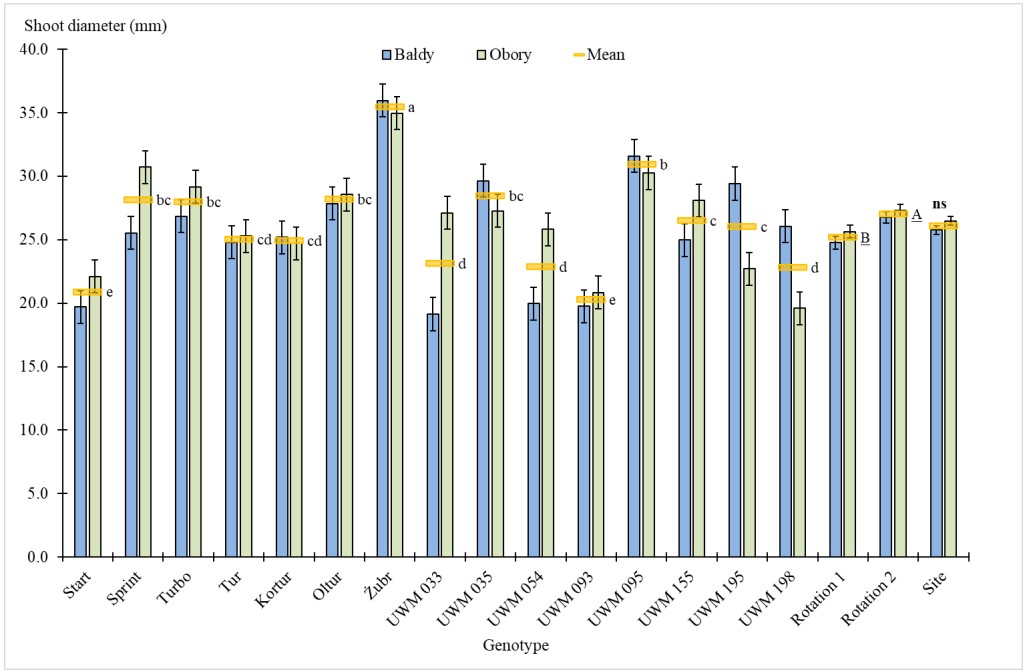

**Figure 4.** The shoot diameter of 15 willow genotypes from two sites in two consecutive three-year rotations (legend: see Figure 1).

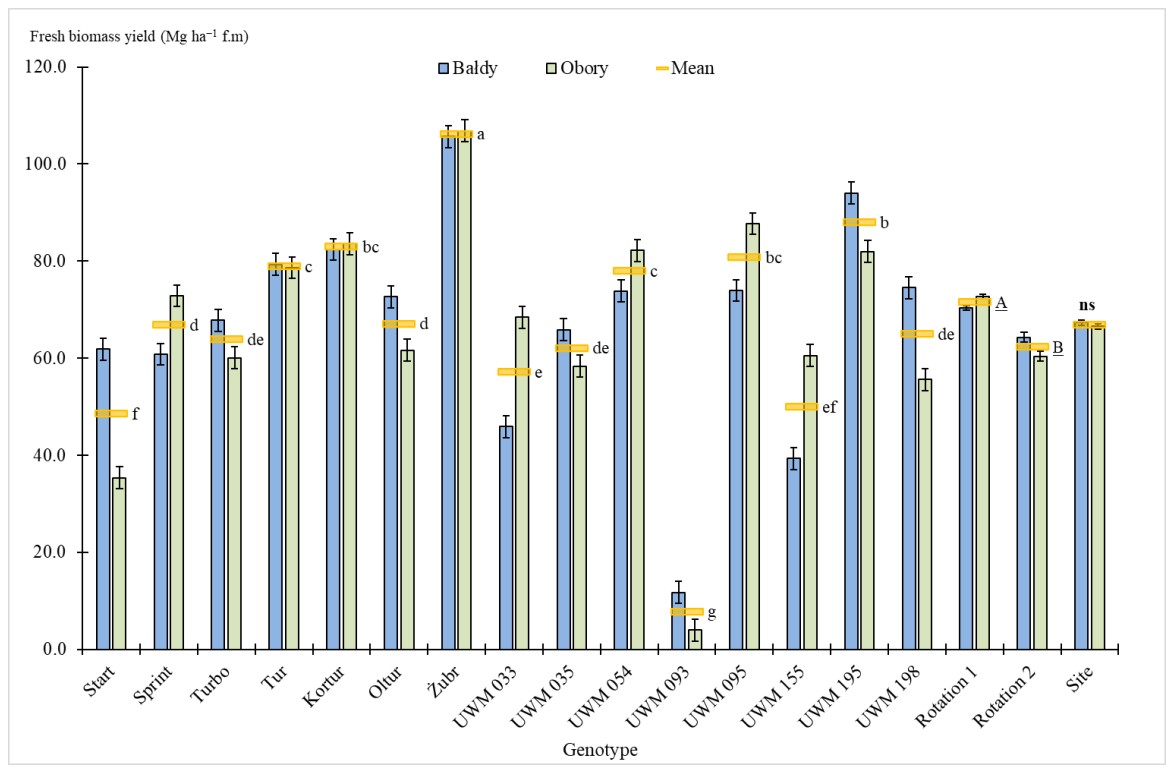

**Figure 5.** Fresh biomass yield of 15 willow genotypes from two sites in two consecutive three-year rotations (legend: see Figure 1).

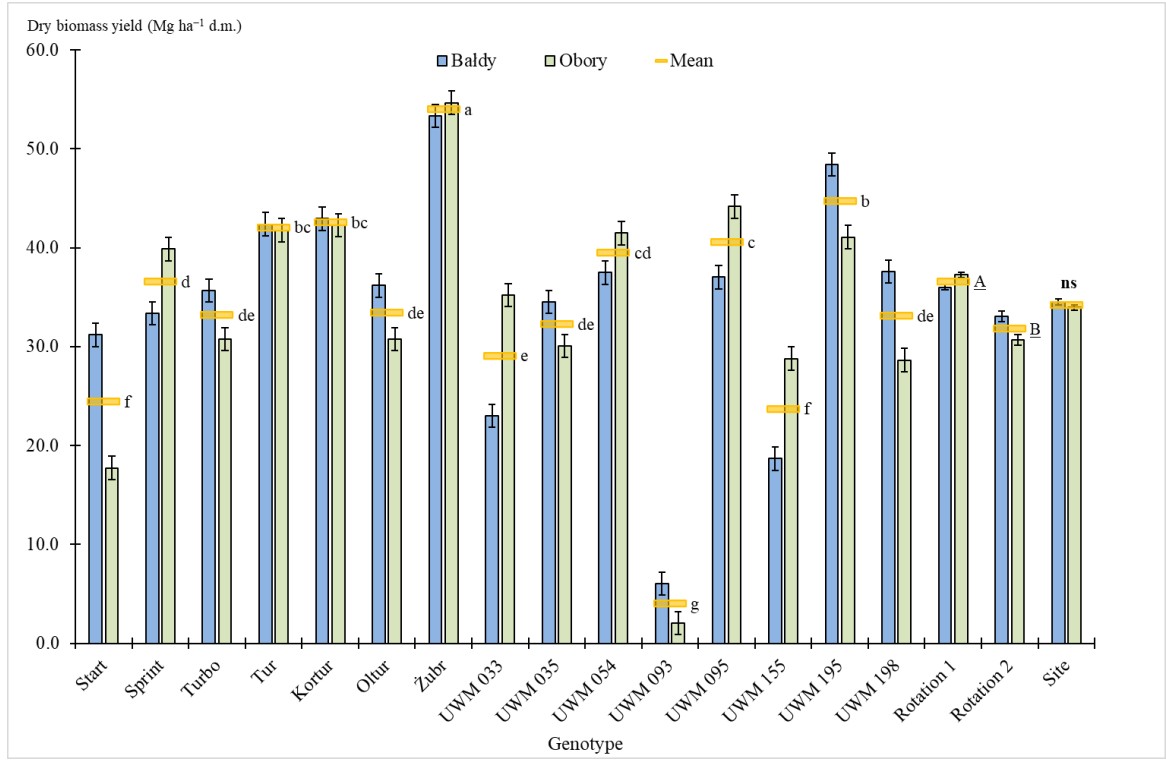

**Figure 6.** Dry biomass yield of 15 willow genotypes from two sites in two consecutive three-year rotations (legend: see Figure 1).

Both the fresh and dry biomass yields were significantly affected by the plant survival rate and their biometric features. The Pearson correlation coefficients (Table 3) showed significant inter-relations between the fresh or dry biomass yield and the plant height, shoot diameter and the plant survival rate. However, for the latter feature, the authors considered the number of plants, which survived until the subsequent 3-year harvest rotation rather than the initial planting density.

The greatest correlation strength was observed for the plant height (up to 0.76). It is this feature that was the principal determinant of the biomass yield, although the number of shoots did not correlate significantly with the biomass yield. Multiple regression models (Table 4) have confirmed these relations and the $R^2$ coefficients not only expand interpretations by providing information on a good fit of the regression models to experimental data, but also these three significant characteristics alone explained 75% of the biomass yield variance.

**Table 3.** Pearson correlation coefficients between the fresh and dry biomass yield and the yield structure elements: survival rate, number of shoots, height and diameter.

| Variable | Survival | No. of Shoots | Height | Shoot Diameter |
|---|---|---|---|---|
| Fresh matter yield | 0.57 ** | 0.15 | 0.76 ** | 0.55 ** |
| Dry matter yield | 0.56 ** | 0.17 | 0.75 ** | 0.54 ** |

$** p < 0.01.$

**Table 4.** Multiple regression analysis in an assessment of the relationship between fresh or dry matter yield and biometric features.

| Parameter | Parameter Value | *p*-Value |
|---|---|---|
| | Fresh matter yield | |
| Intercept | **−98.95** | **<0.0001** |
| Survival | **2.18** | **<0.0001** |
| No. of shoots | 0.85 | 0.4826 |
| Height | **15.12** | **<0.0001** |
| Shoot diameter | **1.21** | **0.0186** |
| $R^2$ | 0.76 | |
| $R^2_{adj.}$ | 0.74 | |
| | Dry matter yield | |
| Intercept | **−51.27** | **<0.0001** |
| Survival | **1.11** | **<0.0001** |
| No. of shoots | 0.58 | 0.3615 |
| Height | **7.64** | **0.0001** |
| Shoot diameter | **0.64** | **0.0180** |
| $R^2$ | 0.75 | |
| $R^2_{adj.}$ | 0.73 | |

Significant parameters are shown in bold.

The coefficients of variation analyses showed the lowest variation (13.3%) in the total approach for height and the highest (38.3%) for the number of shoots (Table 5). The same pattern was observed when this variation was broken down into the principal components: site, genotype and rotation. The biomass yield variation was 36%. Therefore, since biomass yield depends on several uncontrollable biotic and abiotic factors, it can be claimed that these results are good and relatively stable. Moreover, taking into account the effect of the principal components on the variation of the characteristics, the lowest variation was determined by the genetic factor (genotype). The genotype variation ranged from 5.9% (height) to 30.7% (number of shoots) and the effect of the other two principal factors (site and rotation) had a similar impact on individual variables.

It is very important in an assessment of the willow biomass yield to present the dry biomass yield per one year of plantation use. Such yield in the whole experiment presented in this paper amounted

to 11.4 Mg ha$^{-1}$ year$^{-1}$ d.m., regardless of the genotype, site or harvest rotation (Figure 7). Moreover, seven out of the 15 genotypes under study gave a higher yield than the mean from the whole experiment. However, the differences between genotypes, sites and rotations were large because the highest-yielding Żubr variety gave a mean yield of 18 Mg ha$^{-1}$ year$^{-1}$ d.m., and 20.3 Mg ha$^{-1}$ year$^{-1}$ d.m. in the first harvest rotation at Obory (Table 6). The mean yield for the best three and five genotypes was lower by 13% and 17%, respectively, and the mean yield of the whole experiment was lower by 37% compared to the mean yield of the best variety (Żubr). It is also noteworthy that all of the genotypes in the second harvest rotation at each site gave a lower yield than in the first rotation. The yield decrease in the second rotation at Obory was much larger (mean 18%) compared to Bałdy (mean 8%). Moreover, only in the first rotation was the mean dry biomass yield higher at both sites than the mean for the whole experiment.

**Table 5.** Percent coefficients of variation (CV%) for site, genotype, harvest rotation and total for the willow characteristics under study.

| Variable | Site | Genotype | Rotation | Total |
|---|---|---|---|---|
| Survival | 25.42 | 17.43 | 23.59 | 25.50 |
| No. of shoots | 34.95 | 30.73 | 33.10 | 38.31 |
| Height | 13.24 | 5.90 | 13.27 | 13.32 |
| Shoot diameter | 20.25 | 14.25 | 20.01 | 20.24 |
| Fresh matter yield | 35.89 | 18.61 | 35.19 | 35.82 |
| Dry matter yield | 36.27 | 18.97 | 35.59 | 36.19 |

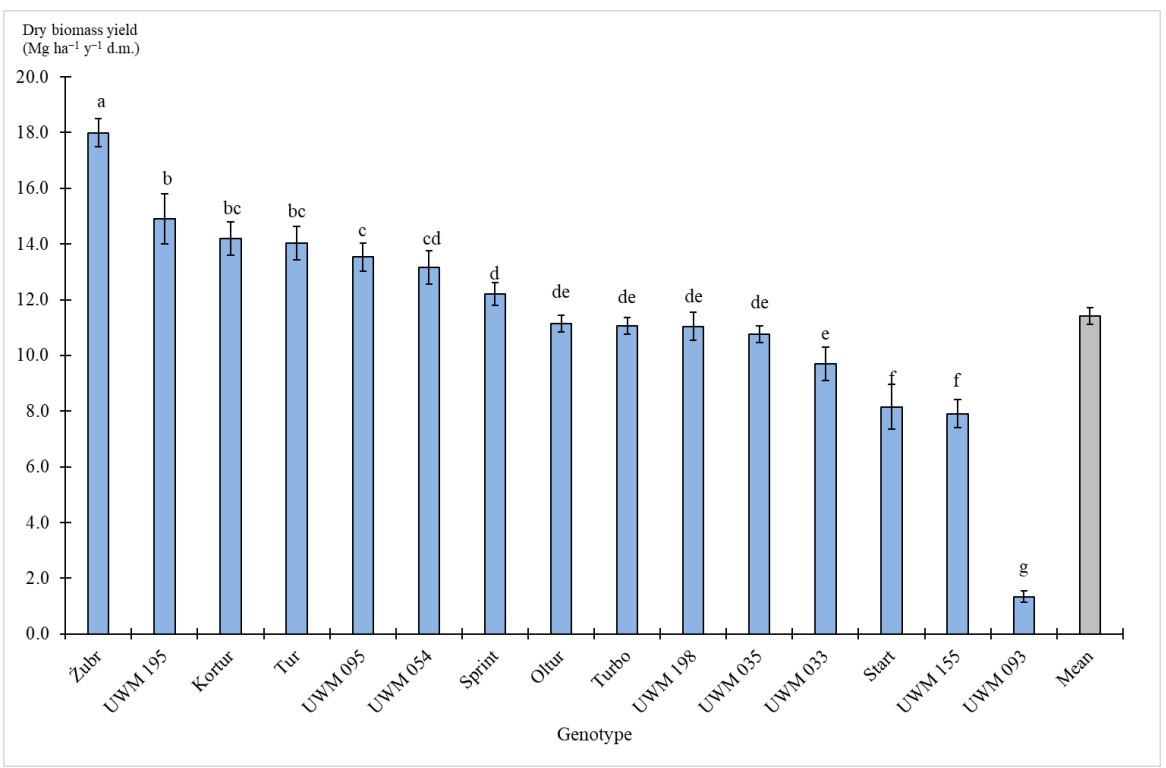

**Figure 7.** Ranking of the mean dry biomass yield (Mg ha$^{-1}$ year$^{-1}$ d.m.) for 15 willow genotypes from two sites in two consecutive three-year rotations (error bars denote the standard error of mean; letters a–f denote homogeneous groups).

**Table 6.** First and second three-year rotation dry biomass yields (Mg ha$^{-1}$ year$^{-1}$ d.m.) and changes in yield for the top 1, 3, 5 and all 15 willow genotypes at two sites in northern Poland.

| Genotype | | Bałdy | Obory | Mean for Two Sites |
|---|---|---|---|---|
| Top 1 | 1st rotation yield | 18.0 | 20.3 | 19.2 |
| | 2nd rotation yield | 17.6 | 16.1 | 16.8 |
| | yield difference | −0.4 | −4.3 | −2.3 |
| | % of yield decrease | 2.1 | 20.9 | 12.1 |
| | mean yield from two rotations | 17.8 | 18.2 | 18.0 |
| | % of mean yield compared to the highest-yielding genotype | 100.0 | 100.0 | 100.0 |
| Top 3 | 1st rotation yield | 16.7 | 17.6 | 17.1 |
| | 2nd rotation yield | 15.4 | 13.1 | 14.3 |
| | yield difference | −1.3 | −4.4 | −2.9 |
| | % of yield decrease | 7.8 | 25.3 | 16.8 |
| | mean yield from two rotations | 16.1 | 15.3 | 15.7 |
| | % of mean yield compared to the highest-yielding genotype | 90.4 | 84.2 | 87.3 |
| Top 5 | 1st rotation yield | 15.6 | 17.0 | 16.3 |
| | 2nd rotation yield | 14.3 | 12.9 | 13.6 |
| | yield difference | −1.3 | −4.1 | −2.7 |
| | % of yield decrease | 8.5 | 24.0 | 16.5 |
| | mean yield from two rotations | 14.9 | 14.9 | 14.9 |
| | % of mean yield compared to the highest-yielding genotype | 84.0 | 81.9 | 83.0 |
| Mean for all 15 genotypes | 1st rotation yield | 12.0 | 12.4 | 12.2 |
| | 2nd rotation yield | 11.0 | 10.2 | 10.6 |
| | yield difference | −1.0 | −2.2 | −1.6 |
| | % of yield decrease | 8.2 | 17.7 | 13.0 |
| | mean yield from two rotations | 11.5 | 11.3 | 11.4 |
| | % of mean yield compared to the highest-yielding genotype | 64.7 | 62.1 | 63.4 |

## 4. Discussion

In the present study, the willow plant survival rate decreased naturally with increasing plantation age. It was also found in other research by the authors that the willow survival rate over 12 consecutive years was significantly differentiated by the genotype, harvest rotation and the initial planting density [31]. The willow survival rate in that research decreased rapidly in the first and second harvest rotation (88% and 58%). It was more stable in the third and fourth rotation, with a slightly decreasing trend (51% and 47%). The highest survival rate in the four three-year harvest rotations was determined for the UWM 095 genotype in the first and fourth rotation (91% and 53%, respectively).

The mean survival rate of 30 willow genotypes in research conducted in Canada was also highly varied [35]. The mean willow survival rate after the second three-year rotation in that research was 60% (ranging from 3% to 94%) and it was lower by 25% compared to the first rotation. Furthermore, the survival rate of 12 willow genotypes in the first three-year harvest rotation in a study conducted in the USA ranged from 63% to 100%, and from 60% to 97% in the subsequent rotation [36]. In another

study, the mean survival rate of seven willow genotypes after the first three-year rotation was 75%, but the inter-genotype differences were large, ranging from 44% to 85% [37].

In the present study, the number of shoots per rootstock ranged widely from 1.8 to 6.6, whereas in research on 25 willow plantations in Denmark, the number of shoots also ranged widely—from 1.4 to 9.9 shoots per stool depending on the cultivar, plantation age and harvest rotation [38]. The number of three-year-old willow shoots per stool ranged from 2.1 to 3.1. A similar number of shoots per stool (2.7) for seven willow genotypes in a three-year harvest rotation were found in another study [37]. An even smaller mean number of shoots per stool (1.4–2.3) were found in the willow harvested in the first five-year harvest rotation in Spain [39].

In the present study, the Żubr variety was particularly distinctive since it produced the tallest plants with the largest shoot diameters. Moreover, this plant's survivability was slightly higher, although the number of shoots was lower than the average for all the 15 genotypes. In other studies, three-year-old plants of the Żubr variety (formerly UWM 006) were also the tallest (7.28 m) and the thickest (48.6 mm) [37]. Willows were found to be shorter in the research conducted in Canada, where the mean height for 30 genotypes was found to be 2.55 m, with the best three genotypes being 3.76 m tall [35]. The shoot diameter for 30 genotypes in Canada measured only 11.9 mm and 15 mm for the best three genotypes, which was markedly lower than in the present study.

Willow harvest in three-year rotations can give a higher dry biomass yield compared to the yield in shorter one- or two-year rotations [4,31]. Moreover, the willow biomass yield in the first three-year rotation is generally lower compared to subsequent three-year harvest rotations [13,21,36,40,41]. The yield increase in the second and subsequent harvest rotations was particularly manifest in plantations that were not coppiced in the first growing year to increase the number of shoots per stool. However, in some cases, the biomass yield during the first three-year harvest rotation was much higher than in the subsequent harvest rotations [9,31,42,43].

It must be stressed that the current study dealt with two consecutive three-year harvest rotations and one cannot (and must not) regard them as the first and the second three-year harvest rotation in a direct meaning of the word. This is a consequence of the fact that plants in the experiment were obtained earlier in one-year harvest rotations, and a change of the harvest rotation to three years took place in the fifth and the sixth year of the plantation use, when the plants' root systems were strong (which is also explained in the Materials and Methods section). Plants harvested in the first three-year rotation grew on eight-and seven-year-old stools, so the root systems can be regarded as very well-developed and the plants were giving the full yield. Nevertheless, the study showed that the duration of a willow harvest rotation can be changed. This knowledge is also important to potential investors, as it gives them, in a sense, some flexibility in choosing the willow harvest rotation.

The mean biomass yield from two three-year harvest rotations in this study for most of the 15 genotypes under study (except one) ranged from 8 to 18 Mg ha$^{-1}$ year$^{-1}$ d.m., with the mean yield being 11.4 Mg ha$^{-1}$ year$^{-1}$ d.m. However, because of the plantation (and the plants' root systems) age, the biomass yield obtained in the current experiment should be compared with the findings of other experiments, starting with the second three-year harvest rotation. For this reason, literature data were used and Figure 8 shows a comparison of the willow biomass yield mainly in the second and the third harvest rotation [9,13,21,31,35,36,38,40,41,44–52], although the harvest rotation was shorter or longer by one year in some studies. Since the studies were conducted in different countries, the site conditions (soil and weather) were clearly different. Moreover, the plantations were set up with various clones and various agrotechnical and plantation management procedures were applied. Nevertheless, the findings of this study concerning the mean dry biomass yield in the three-year harvest rotation were on a medium or high level compared to the other studies cited in Figure 8. Moreover, the current results were similar to those of a study conducted at five different sites in the USA (mean range 10.0–11.6 Mg ha$^{-1}$ year$^{-1}$ d.m.) [9].

In the present study, the best-yielding variety—Żubr—gave a higher biomass yield at both sites and harvest rotations (range: 16–20 Mg ha$^{-1}$ year$^{-1}$ d.m.) compared to the levels achieved for the best genotypes in the USA (range 11–14 Mg ha$^{-1}$ year$^{-1}$ d.m.) [9]. A large increase in the biomass yield in the

second three-year harvest rotation was also achieved in Canada, mean 13.3 Mg ha$^{-1}$ year$^{-1}$ d.m. [44]. Similar mean yield as in Canada was obtained in the authors' earlier research of five willow genotypes grown at four planting densities and harvested in four three-year rotations [31]. The highest mean yield (14.5 Mg ha$^{-1}$ year$^{-1}$ d.m.) was achieved in the cited experiment from the UWM 095 genotype, i.e., it was higher only by one Mg ha$^{-1}$ year$^{-1}$ d.m. compared to the mean value for this genotype achieved in the authors' research. The other genotypes gave lower yields of 2.3%, 5.1%, 8.2% and 24.7%, respectively, compared to the UWM 095 genotype [31]. However, one must add that the willow yield in four consecutive three-year harvest rotations, depending on the experimental factors as determined in the cited research, varied strongly and ranged from 4.8 to 23.2 Mg ha$^{-1}$ year$^{-1}$ d.m.

Considerable diversity (from 3.5 to 13.6 Mg ha$^{-1}$ year$^{-1}$ d.m.) was also shown in other research conducted in the USA using 18 willow genotypes at two sites in a three-year harvest rotation [23]. Furthermore, in a study conducted in Sweden, the mean yield for two willow cultivars grown at five sites with no fertilisation amounted to 5.9 Mg ha$^{-1}$ year$^{-1}$ d.m. [17]. The highest yield (13.2 Mg ha$^{-1}$ year$^{-1}$) was achieved with intensive fertilisation. The willow yield obtained in Denmark also ranged widely from 2.4 to 15.1 Mg ha$^{-1}$ year$^{-1}$ d.m. [38]. Moreover, as in other studies, the yield in the second rotation was higher than in the first one.

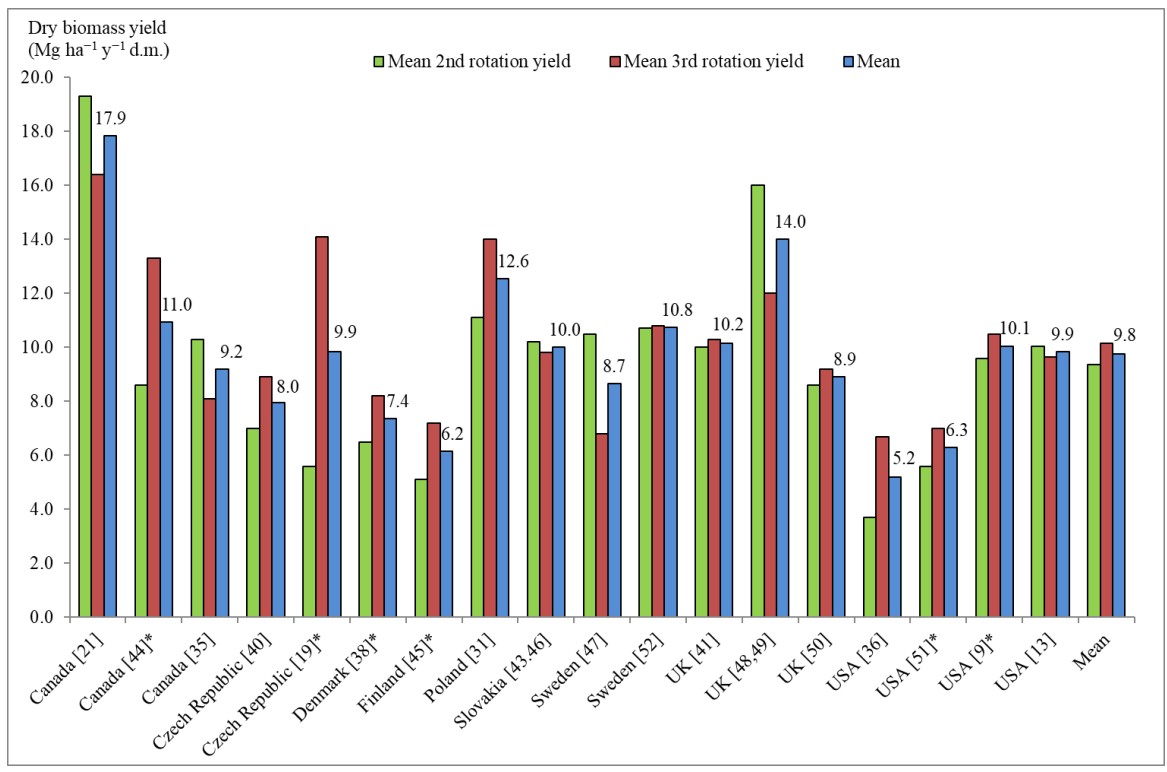

**Figure 8.** A comparison based on literature data of willow biomass yield (Mg ha$^{-1}$ year$^{-1}$ d.m.) in the second and the third harvest rotation, although the harvest rotation was shorter or longer by one year in some studies (* data from the first and second three-year willow harvest rotation).

## 5. Conclusions

The current study analysed 15 genotypes grown at two different sites and harvested in two consecutive three-year harvest rotations and demonstrated the very high impact of the genotype (81%) on the yield of willow grown as SRWC. The harvest rotation, along with the genotype, had a significant impact on the plant survival rate and the number of shoots per stool. The results showed that the willow biomass was mainly affected by the plant height, survival rate and shoot diameter. The differences in the biomass yield between the genotypes under study were very large, and the Żubr variety of species *S. viminalis* gave a particularly high yield. The mean yield for the best three and five

genotypes was 13% and 17% lower, respectively, and the mean yield for the whole experiment was 37% lower compared to the mean yield of the best variety (Żubr). Therefore, the choice of a willow genotype is of key importance to success in SRWC willow production, since single-genotype monoculture on a commercial plantation may be a significant source of future problems with disease development or pest infestations. Therefore, further research is needed in subsequent harvest rotations to verify the yielding stability and to assess the economic, energy and environmental viability of biomass production based on the highest-yielding genotypes. Such knowledge will improve production and increase interest among producers and consumers of this renewable biomaterial.

**Supplementary Materials:** The following are available online at http://www.mdpi.com/2077-0472/10/9/0412/s1, Figure S1: Monthly precipitation (mm) in the years of the experiment at two sites Bałdy (a) and Obory (b) and mean value for years 2013–2018., Table S1: Mean monthly air temperature (°C) in the years of the experiment at two sites (Bałdy and Obory) and average temperatures in the growing seasons (IV-X) and throughout the whole year.

**Author Contributions:** Conceptualisation, M.J.S., S.S., J.T.; Data curation, M.J.S., M.K.; Formal analysis, M.J.S., M.K., D.Z.; Funding acquisition, M.J.S., S.S.; Investigation, M.J.S., S.S. and M.K.; Methodology, M.J.S., S.S., J.T. and M.K.; Project administration, M.J.S.; Validation, M.J.S., M.K., and D.Z.; Visualisation, M.J.S., D.Z.; Writing—original draft, M.J.S.; Writing—review and editing, M.J.S., M.K., D.Z., J.T. and S.S. All authors have read and agreed to the published version of the manuscript.

**Funding:** This paper is the result of a long-term study carried out at the University of Warmia and Mazury in Olsztyn, Faculty of Environmental Management and Agriculture, Department of Plant Breeding and Seed Production, topic number 20.610.008–300 and it was co-financed by the National (Polish) Centre for Research and Development (NCBiR), entitled "Environment, agriculture and forestry", project: BIOproducts from lignocellulosic biomass derived from MArginal land to fill the Gap In Current national bioeconomy, No. BIOSTRATEG3/344253/2/NCBR/2017.

**Acknowledgments:** We would also like to thank the staff of the Department of Plant Breeding and Seed Production for their technical support during the experiments.

**Conflicts of Interest:** The authors declare no conflict of interest.

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
