# Peer review of "Effects of Site, Genotype and Subsequent Harvest Rotation on Willow Productivity"

_agriculture, doi:10.3390/agriculture10090412_

Round 1
Reviewer 1 Report
Ms needs considerable editing throughout, such as:
- Title could be Effects of Site, Genotype, and Rotation Length on Willow Productivity in Poland
- Authors addresses are needed in line 8.
- Deleting "It is noteworthy that" preceding "willow" in line 42.
- UWM could be defined for "University of Warmia and Mazury" in line 7.
- #s less than 10 should be spelled out in text.
- Table 2 and Figure 1 could be deleted.
- Consolidate Figures 2, 3, 4, 5, and 7 into one table with traits as columns and genotypes as rows; delete Figure 6.
- "Steam" in Tables 4, 5, and 6 should be "stem".
- Delete "It has been shown in many studies that" from beginning of line 315.
- Delete Table 7 and incorporate its important points in text.
- Delete "It must be stressed that" at beginning of line 325.
- References are not uniformly formatted.
Line 75 needs to have Poland added.
What is the total area of each planting?
What is the plot size (line 107)?
What are the harvest dates?
Author Response
Response to Reviewer 1 Comments
Manuscript ID: agriculture-905240
Title: Contributions of plantation localisation, genotype and harvest rotation on willow biometric features and biomass yield
Ms needs considerable editing throughout, such as:
- Title could be Effects of Site, Genotype, and Rotation Length on Willow Productivity in Poland
As suggested by Reviewer 1 and Reviewer 2, the title was changed to:
Effects of Site, Genotype and Subsequent Harvest Rotation on Willow Productivity
The language of the paper was revised by native speaker.
- Authors addresses are needed in line 8.
The addresses have been added.
- Deleting "It is noteworthy that" preceding "willow" in line 42.
Corrected as suggested by Reviewer 1.
- UWM could be defined for "University of Warmia and Mazury" in line 7.
Corrected as suggested by Reviewer 1.
- #s less than 10 should be spelled out in text.
Corrected as suggested by Reviewer 1.
- Table 2 and Figure 1 could be deleted.
As suggested, Table 2 and Figure 1 were deleted from the main manuscript, but they were included in a separate file as supplementary materials.
- Consolidate Figures 2, 3, 4, 5, and 7 into one table with traits as columns and genotypes as rows; delete Figure 6.
In our opinion, merging so many figures to make one large table (with 54 rows) will make the data unclear and unattractive to the readers. In our opinion, presenting them in figures is of higher value as one picture is often more meaningful than a lot of numbers. Therefore, we suggest that the figures should be left unchanged. The same applies to Fig. 6. We realize that scientific studies usually use dry weight of yield for comparisons. However, fresh biomass is always harvested on a plantation, so the information on the fresh biomass yield is also important from the practical perspective and it affects production effectiveness and profitability. Therefore, we are of the opinion that the information presented in Fig. 6 is important and we suggest that it should be left unchanged.
- "Steam" in Tables 4, 5, and 6 should be "stem".
Changed to “shoot” throughout the paper.
- Delete "It has been shown in many studies that" from beginning of line 315.
Corrected as suggested by Reviewer 1.
- Delete Table 7 and incorporate its important points in text.
We are of the opinion that Table 7 (currently Table 6) presents important information, which is why we suggest that it should be left unchanged.
- Delete "It must be stressed that" at beginning of line 325.
We suggest leaving this part of the sentence unchanged as it is very important in the context of our research.
- References are not uniformly formatted.
References have been corrected.
Line 75 needs to have Poland added.
“Poland” added as suggested by Reviewer 1.
What is the total area of each planting?
The information was added in Materials and Methods.
What is the plot size (line 107)?
The information was added in Materials and Methods.
What are the harvest dates?
The information was added in Materials and Methods.

Reviewer 2 Report
Dear Authors,
Please find the comments in the attached file.
Regards.

Author Response
Response to Reviewer 2 Comments
Manuscript ID: agriculture-905240
Title: Contributions of plantation localisation, genotype and harvest rotation on willow biometric features and biomass yield
L 2-3 somehow subsequent rotations should be highlighted
As suggested by Reviewer 1 and Reviewer 2, the title was changed to:
Effects of Site, Genotype and Subsequent Harvest Rotation on Willow Productivity
L 13: I think here is better not to use discussive sentence or be more straight forvard " provides...."
Corrected as suggested by Reviewer 2.
L 17: 81% of what?
It has been explained that it was about 81% of the total variance.
L 20: dont use this word if you did not do statistical analysis. Instead you can use word remarkably for example.
Obviously, the statistical analysis was performed, which is mentioned in the Materials and Methods and in the Results. Therefore, we can, and should, use the word “significantly”.
L 72: This section is not clear and has to be inproved. For each task in the introduction, method has to be defined clearly.
The Materials and Methods section has been supplemented and corrected. However, we do not understand why this section has to address each task in the introduction. The Materials and Methods section presents our experiment and the method of carrying it out so that every reader can reproduce it, and we think it is sufficient. Moreover, this approach is widely applied in scientific papers.
L 84 This is very important point that has to be further discussed in the discussion
We agree with this suggestion and it has been done.
L 94 Put it to the discussion part.
We included this information in the Discussion section, but we also left it in the Materials and Methods, so that a manuscript reader can understand fully how our experiment was conducted.
L 106 Please clarify which biometric features?
The relevant information was added.
L 107 It is not clear is it a sample plot? or which plot, what size of the plot and so on. Is it for both experiments or 3 plots for each experiment?
The information was added in Materials and Methods.
L 109 Please clarify how many trees were measured in each plot also for each experiment in total. The amount of your measurements is not clear. Also, randomly selected in the field sounds very subjective. How did you implemented it in the field this random? Did you throw a coin to determine if the tree has to be measured or not? Or used some other method?
The information was added in Materials and Methods. Standard methods were applied, which are widely used in field experiments for random selection of experimental objects. No coin tossing, no pseudorandom numbers or other similar methods were used.
L 116 Pleae provide the details in figures. Representative to say it is not enough. How many trees were sampled , from which place samples were taken and so on?
The information was added in Materials and Methods.
L 117-18 Please describe the drying process. Drying temperature and so on.
The information was added in Materials and Methods.
L 120 Please be here more specific with the title by writing task of the study , like contribution or site related factors. In this way it would be more clear, where the statistical analysis is used.
Chapter 2.3. “Statistical analysis” was written in a typical form, which is used in all periodicals included in the Journal Citation Reports, including those published by MDPI. The statistical analyses results are presented in tables (e.g. results of repeated measures analysis of variance) or in graphs (e.g. homogeneous groups in Tukey’s test). The tables and graphs with the statistical analyses results are described so that the reader should have no doubts about what is being presented.
L 131 I see no added value in presenting so much information about the weather (1 table and 2 graphs) and you do not analyze the impact neither of soil (chemistry part) nor weather.
I think you could provide information about the soil and some information about the weather where you are describing the experimental sites. Then this paragraph could be taken out.
In our opinion, the information presented in chapter 3 is important and we suggest that it should be left unchanged. However, as suggested, Table 2 and Figure 1 were deleted from the main manuscript, but were included in a separate file as supplementary materials.
L163 Please separate results part and discussion part. It would be much more easier for the reader to understand and interpret your results. I dont think so that this is an appropriate way to go when writing about the yield studies with a lot of numbers and figures.
Let us point out that a number of papers combine the study findings with their discussion, which sometimes helps to avoid repeating the authors’ own results and returning to an issue which has already been discussed. However, as suggested by Reviewer 2, the results and discussion sections were separated.
L 197 (figure 2). Please describe the Y axis in all your figures.
In all the graphs, the units for the Y axis were given at the top, above the last value on the axis. And the figure subject was described in its title. However, as suggested by Reviewer 2, we added this information in the Y axis description.
L 218 Did you do statistical analysis on this and can provide p value? if no please use some other words like remarkably....
Yes, of course, it was done. The statistical significance of differences is presented here as homogeneous groups in Tukey’s test with p<0.05 (see description for Fig. 1 and the “Statistical analysis” section).
L 227-28 (figure 3). It is hard to understand the meaning of small letters in your graph. Also, if it is very important part of your graph please describe their meaning in methods. If they are not important for your results may be it would be better to take them off. Also their meaning has to be described under each graph where you use them.
The homogeneous groups in Tukey’s test were presented under Fig. 2 (now Fig. 1). We expanded the description for clarity. In order to avoid repeating the same information under consecutive graphs, we suggest referring to the earliest legend, i.e. one provided under Fig. 1.
L 224 I think in the discussion part it is important to discuss that ZUBR had relatively low survival rate and higher diameter. Thus was the most productive. Yet, what could be the result under different densities. Because it seems that 40000 plants/ha is too high density if you do not do cuttings after each year.
It has been emphasized. Obviously, it is clear that 40,000 plants/ha is an excessively high density for willow harvest in three-year rotations. We explained it in the Materials and Methods, and also in the discussion, that initially the willow was harvested every year and the harvest cycle was subsequently extended. The plant density decreased by over 50% compared to the initial value, which is also emphasized in the manuscript.
L 246 I would suggest to use mean annual increment, instead of total biomass. Also you could separate regarding the rotation if possible.
The biomass yield is presented in different ways, also with reference to the annual growth. But let us stress that plants from the plantation are always harvested as fresh biomass in a defined cycle. Therefore, the information on the fresh biomass yield per 1 ha in a three-year cycle is also important from the practical perspective (for farmers and biomass consumers) and it affects the production effectiveness and profitability. Therefore, in our opinion, it is not a problem if we present these figures both as referred to 1 ha and to 1 year of the plantation use.
L 278 please write some words in your methods about it.
The description was added to “Statistical analysis”.
L 282 I Doubt if you have linear relations between these variables analyzed. Have you checket it?
We wish to explain that our statistical analyses confirmed the existence of a strong correlation between the yield and the plant height, which should raise no doubts. Our study also found a linear correlation between the biomass yield and shoot diameter, although the correlation was weaker than between the yield and height. However, there was certainly no linear relationship between the yield and the number of shoots, which was demonstrated both by the correlation and by multiple regression through the absence of statistical significance. Nevertheless, we left this statistically non-significant effect in the regression model to show the most important features of the yield structure. It might seem that the greatest doubts as to the linear relationship can be raised by the survival rate. However, let us stress clearly that this does not concern the initial planting density, but the number of plants that survived in a three-year harvest rotation.
L 284 Are you sure about the linear relations. Proper description in the methods part is needed.
This is explained above and the description was included in “Statistical analysis”.
L 308 No single word in methods part. Please describe it where properly.
The description was added to “Statistical analysis”.
L 312 No single word about the ranking and its purpose in the methods part. Please describe it.
The description was included in Materials and Methods.
L 366 (figure 9) I think in the discussion part it would be enought to leave the text (lines 340-363 or other needed. This figure is much too much. I would sugest to take it out.
We do not agree. Obviously, there are certain elements in the discussion that describe Fig. 9, but describing all the data in sequence seems unnecessary, it will be unclear and may be boring to the reader. However, in our opinion, presenting various findings in one figure is of higher value as one image is often more meaningful than a lot of text and numbers. Therefore, we suggest that this figure (currently Fig. 8) should be left unchanged.
L 379-84 This belongs to discussion part
In our opinion, this part can also be left in the conclusion of our research as it shows the direction of the future research of the subject. Such an approach, in which the conclusion also identifies the direction of potential future research, is commonly applied in scientific papers.

Round 2
Reviewer 1 Report
While the ms has been improved considerably, numerous opportunities for concise writing and consistent formatting of citations remain.
Suggestions include:
Line 18 - delete "The analysis showed that the"
Line 19 - delete "Among the genotypes under study"
Line 33 - delete ", which yield woody biomass"
and similar edits of Lines 42-43, 79, 206, 256, 324, 339, 389.
Remove title word caps in citations 9,13, 17,18, 23, 28,30,34,38, 39.
Since fresh weight results are basically the same as dry weights, suggest deleting Figure 5 and fresh weights in Tables 3, 4, and 5
Author Response
While the ms has been improved considerably, numerous opportunities for concise writing and consistent formatting of citations remain.
Suggestions include:
Line 18 - delete "The analysis showed that the"
Corrected as suggested by Reviewer 1.
Line 19 - delete "Among the genotypes under study"
Corrected as suggested by Reviewer 1.
Line 33 - delete ", which yield woody biomass"
Corrected as suggested by Reviewer 1.
and similar edits of Lines 42-43, 79, 206, 256, 324, 339, 389.
Corrected as suggested by Reviewer 1.
Remove title word caps in citations 9,13, 17,18, 23, 28,30,34,38, 39.
Corrected as suggested by Reviewer 1.
Since fresh weight results are basically the same as dry weights, suggest deleting Figure 5 and fresh weights in Tables 3, 4, and 5.
We realize that scientific studies usually use dry weight of yield for comparisons. However, fresh biomass is always harvested on a plantation, so the information on the fresh biomass yield is also important from the practical perspective and it affects production effectiveness and profitability. Therefore, we are of the opinion that the information presented in Fig. 5 and in Tables 3, 4, 5 are important and we suggest that it should be left unchanged.
